# Reproducibility Study of "XRec: Large Language Models for Explainable Recommendation"

## Abstract

In this study, we reproduced the work done in the paper "XRec: Large Language Models for Explainable Recommendation" by Ma et al. (2024). The original authors introduced XRec, a model-agnostic collaborative instruction-tuning framework that enables large language models (LLMs) to provide users with comprehensive explanations of generated recommendations. Our objective was to replicate the results of the original paper, albeit using Llama 3 as the LLM for evaluation instead of GPT-3.5-turbo. We built on the source code provided by Ma et al. (2024) to achieve our goal. Our work extends the original paper by modifying the input embeddings or deleting the output embeddings of XRec's Mixture of Experts module. Based on our results, XRec effectively generates personalized explanations and its stability is improved by incorporating collaborative information. However, XRec did not consistently outperform all baseline models in every metric. Our extended analysis further highlights the importance of the Mixture of Experts embeddings in shaping the explanation structures, showcasing how collaborative signals interact with language modeling. Through our work, we provide an open-source evaluation implementation that enhances accessibility for researchers and practitioners alike. Our complete code can be found at `https://anonymous.4open.science/r/xrec-repro-C2CD/`.

## 1 Introduction

Recommender systems have proven to be valuable tools for navigating the growing abundance of online content and products, helping with aligning options more closely to users' preferences. Collaborative filtering (CF), a widely used approach in recommender systems, suggests new items based on users' purchase histories and the preferences of others with similar tastes. Traditionally, CF relied on matrix factorization techniques and simpler algorithms to uncover these relationships (Koren & Bell, 2011). However, the rise of deep learning techniques has significantly changed the field of CF, with new architectures like attentive CF (Chen et al., 2017), graph neural networks (GNNs) (He et al., 2020), and self-supervised learning (SSL) (Xia et al., 2023) being significantly better at understanding complex data in order to make more intelligent recommendations while maintaining important patterns in the data. Despite these advancements, many recommender systems continue functioning as black boxes, offering limited interpretability behind the reasoning process for user-item interactions. In light of this, several research studies have focused on introducing more transparency to users regarding recommendations and helping them understand the decision-making process behind the recommendations. Notable works include Att2Seq (Dong et al., 2017) and NRT (Li et al., 2017), which use attention mechanisms and recurrent neural networks (RNNs) to generate textual explanations. Li et al. (2021) and Li et al. (2023) have further utilized the transformer architecture for text generation, providing valuable insights into recommendation results. However, these approaches struggle to create high-quality explanations due to insufficient explanation data.

Building on previous research and utilizing the advanced language capabilities of state-of-the-art LLMs, Ma et al. (2024) introduce XRec, a novel model-agnostic collaborative instruction-tuning framework. XRec enhances LLMs' ability to understand complex user-item interactions by incorporating CF through a unique instruction-tuning method. To bridge the gap between collaborative relationships and language understanding, the authors propose a lightweight collaborative adapter that integrates user behavior signals, enabling

LLMs to better capture user preferences. Additionally, their work addresses the challenge of explainability in recommender systems, providing personalized and interpretable explanations for user-item interactions.

The XRec paper falls under the scope of Transparency within the broader Fairness, Accountability, Confidentiality and Transparency (FACT) topics. Transparency is crucial for AI applications as it helps foster user trust and provide explanations and insights into the decision making process of the otherwise black-box algorithms. Good explanations should be contrastive, selective and socially relevant (Miller, 2019). XRec aligns with these principles by tailoring explanations to specific user-item interactions, and makes them socially relevant by producing natural language output that is easily understandable to users. Moreover, XRec is a post-hoc explainability approach, as it uses collaborative signals and LLMs to analyze and explain model behavior after the recommendations have been generated. This places it among modern efforts to provide local explanations that focus on specific decisions, akin to techniques like counterfactual reasoning or LIME (Ribeiro et al., 2016).

The aim of this paper is to reproduce the results presented by Ma et al. (2024) and further evaluate the XRec framework through two additional experiments. Specifically, we answer the following research questions:

**RQ 1:** To what extent can we reproduce the results of the experiments done by Ma et al. regarding the performance of the XRec framework?

**RQ 2:** What impact does the removal of the adapted user and item embeddings generated by the Mixture of Experts (MoE) have on the generated explanations, in terms of model performance?

**RQ 3:** Do the output embeddings of the GNN provide valuable information to the MoE module for generating better explanations?

The remainder of this paper is organized as follows. Section 2 outlines the scope of our reproducibility study. In Section 3 we provide details on our methodology and experimental setup. Section 4 presents the results of our reproducibility study alongside the two extension experiments. Finally, Section 5 discusses the implications of our findings and concludes the paper.

## 2 Scope of Reproducibility

The authors make several key claims about the framework's performance and capabilities, which we aim to verify in this reproducibility study. These claims include:

**Claim 1: Explainability and stability.** XRec consistently outperforms the baselines in terms of explainability and stability.

**Claim 2: Unique explanations.** XRec generates truly unique explanations for each distinct user-item interaction.

**Claim 3: User and item profiles.** The inclusion of user and item profiles improves the explainability and stability of XRec.

**Claim 4: Collaborative information injection.** The injection of collaborative information improves the explainability and stability of XRec.

Beyond reproducing the original results provided by Ma et al. (2024), we extend their work by further investigating the influence of user and item embeddings generated by the MoE module. Additionally, we investigate the effect of the output embeddings provided from the GNN to the MoE module.

## 3 Methodology

### 3.1 Model Description

The XRec framework, illustrated in Figure 1, unifies graph-based CF with LLMs to provide explainable recommendations. The overall pipeline consists of three main components described below.

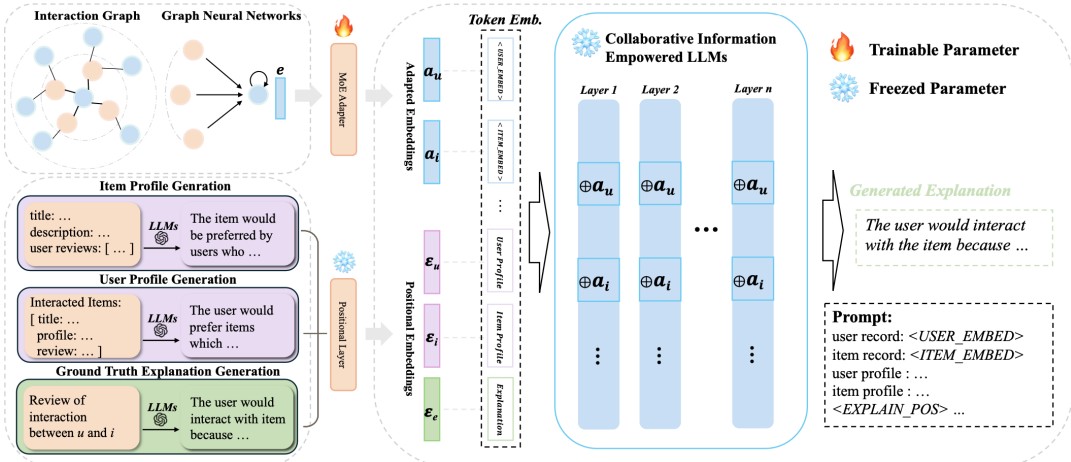

Figure 1: The overall architecture of XRec as shown in the original paper (Ma et al., 2024).

### 3.1.1 Collaborative Relation Tokenizer

XRec begins with a GNN that propagates embeddings among connected nodes in a user–item interaction graph to capture higher-order collaborative relationships. Specifically, LightGCN (He et al., 2020) is adopted for efficient message-passing, averaging information from neighbors at each layer. After $L$ propagation layers, the final user and item embeddings are obtained, encoding their implicit preferences. These embeddings are optimized using a Bayesian Personalized Ranking (BPR) objective, ensuring they align with real user interactions.

### 3.1.2 Collaborative Information Adapter

Although these GNN-derived embeddings capture user–item relationships, they need to be aligned with the token-level representation space of LLMs. To this end, XRec employs an MoE adapter, which transforms the numeric GNN embeddings into "adapted" embeddings that can be inserted into the LLM. Each expert in the MoE is a linear projection, and a learned gating function combines the experts' outputs. This effectively bridges the gap between the collaborative signals and the text-oriented LLM.

### 3.1.3 Unifying CF with an LLM for Explanation Generation

In parallel to generating user and item embeddings, an LLM is prompted to distill concise textual profiles from user interactions (e.g., user reviews) and item descriptions. These textual profiles, along with the adapted GNN embeddings, are then fused by injecting the adapted embeddings into reserved token positions (<USER_EMBED>, <ITEM_EMBED>) at every layer of the LLM.

During training, the LLM's own weights remain frozen, and only the MoE adapter learns to integrate the collaborative signals. The LLM predicts the next token of the explanation, minimizing a negative log-likelihood (NLL) loss over the ground truth text. This procedure enables the model to produce human-readable outputs that reflect both the semantic context (from textual profiles) and the collaborative structure (from the GNN). For a thorough description of each component, please see the original paper by Ma et al. (2024).

## 3.2 Baseline Models

To evaluate the performance of the XRec framework, Ma et al. (2024) compared it to four baseline models. These models are: **Att2Seq** (Dong et al., 2017), an attention-based model that generates reviews using attribute-level information; **NRT** (Li et al., 2017), a multi-task learning framework designed to predict ratings and generate abstractive tips for recommendations; **PETER** (Li et al., 2021), a personalized trans-

former model that maps user and item IDs to textual descriptions for explanation generation; and **PEPLER** (Li et al., 2023), a pretrained transformer incorporating prompt learning strategies to enhance explainable recommendations.

### 3.3 Datasets

To evaluate XRec, the original authors use three publicly available datasets capturing diverse user-item interaction behaviors. **Amazon-books** (Ni et al., 2019) aggregates purchasing behaviors and textual reviews in the books category, comprising 360,839 interactions, 15,349 users, and 15,247 items. **Yelp** (Li et al., 2022) focuses on user-business interactions in the service industry, including ratings and reviews, with 393,680 interactions, 15,942 users, and 14,085 items. Finally, **Google-reviews** (Yan et al., 2023) captures user interactions through Google Maps, incorporating metadata and textual feedback, and accounts for 411,840 interactions, 22,582 users, and 16,557 items.

These datasets are fairly extensive but contain a large number of low-quality reviews. To address this, Ma et al. (2024) applied a $k$-cores algorithm to filter out shorter and less dense reviews, making it easier to generate user and item descriptions (Kong et al., 2019). Additionally, they further reduced the dataset size by subsampling a portion of the data before applying the $k$-cores algorithm. However, since the exact values used in these steps were not disclosed, and the explanation generation for both item and user profiles was conducted using GPT-3.5-turbo, we chose not to regenerate the processed data. The final datasets include Amazon-books (95,841 training, 11,980 validation, and 3,000 test samples), Google-reviews (94,663 training, 11,833 validation, and 3,000 test samples), and Yelp (74,212 training, 9,277 validation, and 3,000 test samples).

### 3.4 Evaluation Metrics

Ma et al. (2024) evaluate the XRec framework using multiple metrics in an effort to address the shortcomings of traditional evaluation methods. While $n$-gram-based metrics like BLEU and ROUGE are widely used, they struggle to capture the semantic meaning of generated explanations. To resolve this, the authors incorporate more advanced metrics—including GPTScore, BERTScore, BARTScore, BLEURT, and the Unique Sentence Ratio (USR)—which better quantify semantic coherence and diversity in explanations.

**GPTScore** evaluates the semantic similarity between a generated explanation and the ground truth by utilizing GPT-based embeddings, aligning closely with human judgment (Wang et al., 2023). Ma et al. (2024) employed GPT-3.5-turbo for this metric. However, since this is a proprietary model, we implemented **LlamaScore**, an alternative to GPTScore which uses 16-bit Llama-3.1-8B-instruct, a model comparable to GPT-3.5-turbo in performance[1]. This provides an open-source solution, making our evaluation framework more accessible to the scientific community while maintaining comparable functionality.

**BERTScore** evaluates token-level cosine similarity using contextual embeddings from BERT (Zhang et al., 2020). **BARTScore** approaches evaluation as a text generation task, with scoring based on the likelihood of regenerating reference texts (Yuan et al., 2021). **BLEURT** enhances generalization and captures subtle semantic nuances by combining pre-training with synthetic data (Sellam et al., 2020), and **USR** assesses diversity by calculating the ratio of unique sentences to total sentences in the generated explanations (Li et al., 2021).

These metrics function as a measure of explainability of the generated explanations. Furthermore, to assess the stability of these explanations, the standard deviation of each metric is calculated, with a lower standard deviation indicating a more stable assessment. We define "performance" as a combination of explainability and stability. This is similar to how Ma et al. (2024) measured performance in their work.

### 3.5 Experimental Setup

Our experiments consist of two main parts: reproducibility and extensions. First, we replicate the results presented in the XRec paper (Ma et al., 2024). We then extend the study with two additional experiments on

---

[1]For a performance comparison between these two models, see `https://ai.meta.com/blog/meta-llama-3-1/`.

the Amazon-books dataset, centered around the roles of the MoE module and the GNN embeddings in the XRec architecture. Both in our replication and extension, some experiments involve omitting certain parts of the XRec architecture. We train these modified architectures from scratch to ensure proper optimization. For all of our experiments, we use the publicly available GitHub repository[2] of the original authors as a starting point. Due to resource constraints, part of the experiments are evaluated on a 10% subset ($N = 300$) of the original testing datasets.

### 3.5.1 Reproducibility Experiments

Our reproduction of the results from Ma et al. (2024) strictly adhered to the original experimental setup. We utilize the provided pre-trained LightGCN embeddings and pre-generated user and item profiles from the authors' codebase without modification, ensuring that only the MoE module is retrained. A total of eight experiments are conducted to validate the original findings, replicating the original author's non-ablation and ablation studies. For the non-ablation studies, we evaluate two configurations: (1) the full XRec model with LightGCN embeddings and generated profiles, and (2) a variant excluding user and item profile generation. These experiments are executed across the three datasets—Amazon-books, Google-reviews, and Yelp—to verify generalizability. As for our replication of the original paper's ablation studies, we conducted experiments to isolate the contributions of key architectural components. First, we test a configuration where the injection of LightGCN-adapted embeddings into the Llama-2 attention layers is disabled. Second, we evaluate a more reduced variant that excludes both embedding injection and profile generation. These ablations are performed on Amazon-books and Google-reviews, aligning with the scope of the original analysis.

### 3.5.2 Extensions

For our extensions, we conduct two additional experiments on the Amazon-books dataset. The first involves omitting the adapted embeddings generated by the MoE, making the LLM rely solely on the user and item profiles as input. In our second experiment, we omit the predefined LightGCN embeddings, replacing them with a fixed random input embedding to the MoE. Here, the MoE learns an optimal setting of a single set of embeddings globally across the whole dataset, instead of receiving different inputs for each training example. The former extension aims to answer RQ 2: "What impact does the removal of the adapted user and item embeddings generated by the MoE have on the generated explanations, in terms of model performance?". The latter experiment relates to RQ 3: "Do the output embeddings of the GNN provide valuable information to the MoE module for generating better explanations?"

## 3.6 Hyperparameters

For all of our experiments, we fixed the number of epochs and batch size at 1. All other hyperparameters are directly adopted from Ma et al. (2024). The MoE is initialized with 8 experts, a dropout rate of 0.2 and a noise factor of 0.01 for the gating router. The learning rate is set to $10^{-4}$, with a weight decay of $10^{-6}$. Furthermore, we implement an early stopping mechanism for some experiments to reduce computational costs. Given a dataset size of $N$, this mechanism is enabled after $\frac{N}{5}$ samples have been processed during training. The training stops if no improvement in average train loss (ATL) is observed for $\frac{N}{10}$ samples. The ATL is defined using a rolling window with a size of 10, i.e. the last 10 training samples.

## 3.7 Hardware and Environmental Impact

All of our experiments are conducted on the Snellius GPU network nodes of the Dutch National supercomputers, managed by SURF. Most experiments utilize an NVIDIA H100 GPU with 94 GiB of memory at 700 Watt and an AMD EPYC 9334 CPU at 210 Watt. Other experiments are performed on an NVIDIA Multi-Instance GPU (MIG) based on the NVIDIA A100 architecture with 40 GiB of memory at 400 Watt and an Intel Xeon Platinum 8360Y CPU at 250 Watt.

To determine the environmental impact of the training and generation procedure, the timings are recorded and used to calculate the $CO_2$ equivalence. This value is defined as

---

[2]One can view the GitHub repository of the original authors at `https://github.com/HKUDS/XRec`.

$$\text{CO}_2 \text{ equivalence} = CI \times PUE \times P \times t \tag{1}$$

where $CI$ is the carbon intensity, $PUE$ the power usage efficiency, $P$ the power in kW, and $t$ the runtime in hours. We use the most recent carbon intensity measurement of 0.22 kg/kWh (CBS, 2024). Additionally, the PUE is 1.2 for the Snellius GPU network (Groeneveld, 2017). Finally, The total wattage of an H100 node is 0.91 kW and 0.65 kW for an A100 node.

## 4 Results

In this section, we highlight the results of the reproducibility and extension experiments. We address each of our three research questions in a separate subsection. Additionally, we outline the consumed computational resources in our experiments.

### 4.1 Results of the Reproducibility Experiments

This subsection relates to RQ 1: "To what extent can we reproduce the results of the experiments done by Ma et al. (2024) regarding the performance of the XRec framework?" To answer this research question, we conducted two reproducibility experiments. The first reproducibility experiment involves recreating the results of the original unmodified XRec model. For the Amazon-books, Google-reviews and Yelp datasets, the model is trained and evaluated in two configurations: (1) in its regular form, and (2) without the use of item and user profiles ("w/o prof."). These results (marked with [R]), alongside the original XRec results (marked with [O]) and the baseline model results provided by Ma et al. (2024), are summarized in Table 1.

In the second reproducibility experiment, we replicate the ablation study results presented in Figure 3 of Ma et al. (2024). For this study, two models are trained and evaluated on the Amazon-books and Google-reviews datasets in two separate experiments: (1) without injecting the adapted embeddings into the LLM layers ("w/o inj."), and (2) without both user profiles and injection ("w/o prof. & inj."). The original results of Ma et al. (2024) and of our experiment are shown in Figure 2.

Using Table 1 and Figure 2, we evaluate the claims made by Ma et al. (2024), mentioned in Section 2.

**Claim 1: Explainability and stability.** The first claim relates to XRec's performance in comparison to baseline models in terms of explainability and stability. Ma et al. (2024) define explainability as the base evaluation metrics and stability as the standard deviation of these metrics, where a lower standard deviation indicates a higher stability, as detailed in Section 3.4. Table 1 shows that alignment between the original and reproduced results varies across metrics and datasets, with our reproduction generally yielding lower mean scores and higher standard deviations in multiple metrics.

Furthermore, Table 1 shows some examples of baseline models outperforming both the original XRec by Ma et al. (2024) and our reproduction results. Testing on the Amazon-books dataset, the PETER model scores higher than the XRec original and reproduction in terms of $\text{BERT}^P$. Additionally, PETER surpasses the reproduced models in $\text{BERT}^{F1}$ performance. For the Yelp dataset, PEPLER outperforms our XRec reproduction in $\text{BERT}^R$. In testing on the Google-reviews dataset, PEPLER's BLEURT scores outperformed our XRec reproduction (both with and without profiles). Overall, the results demonstrate that our reproduction experiments for the XRec framework, both with and without profiles ("w/o prof."), do not consistently outperform or show significant superiority over the baseline models. Due to this, we reject the first claim.

**Claim 2: Unique explanations.** The authors' second claim states that the XRec framework produces truly unique explanations for each distinct item-user interaction. This claim is based solely on the USR, described in Section 3.4. The results of our reproducibility studies, presented in Table 1, show that both the original XRec framework and its variant without user and item profiles ("w/o prof.") achieve a USR of 1.0 across the Amazon-books, Google-reviews, and Yelp datasets. This means all generated explanations were unique. Therefore, we validate claim 2.

**Claim 3: User and item profiles.** The claim states that XRec's performance in explainability and stability metrics is improved by the inclusion of the user and item profiles in the LLM prompt inside of

Table 1: A comparison between the baseline models, the original XRec results and our reproducibility results, in terms of explainability and stability. The superscripts $P$, $R$, and $F1$ indicate precision, recall, and F1-score, respectively. The subscript $std$ denotes the standard deviation of each score. Models marked with † were evaluated using 10% of the test set. The best result for each metric per dataset is highlighted in bold, the second best is underlined.

| Metrics | Explainability ↑ | | | | | | | | Stability ↓ | | | | | | |
|---|---|---|---|---|---|---|---|---|---|---|---|---|---|---|---|
| | GPT | Llama | $BERT^P$ | $BERT^R$ | $BERT^{F1}$ | BART | BLEURT | USR | $GPT_{std}$ | $Llama_{std}$ | $BERT^P_{std}$ | $BERT^R_{std}$ | $BERT^{F1}_{std}$ | $BART_{std}$ | $BLEURT_{std}$ |
| Amazon-books | | | | | | | | | | | | | | | |
| Att2Seq | 76.08 | N/A | 0.3746 | 0.3624 | 0.3687 | -3.9440 | -0.3302 | 0.7757 | 12.56 | N/A | 0.1691 | 0.1051 | 0.1275 | 0.5080 | 0.299 |
| NRT | 75.63 | N/A | 0.3444 | 0.3440 | 0.3443 | -3.9806 | -0.4073 | 0.5413 | 12.82 | N/A | 0.1804 | 0.1035 | 0.1321 | 0.5101 | 0.3104 |
| PETER | 77.65 | N/A | **0.4279** | 0.3799 | 0.4043 | -3.8968 | -0.2937 | 0.8480 | 11.21 | N/A | 0.1334 | 0.1035 | 0.1098 | 0.5144 | 0.2667 |
| PEPLER | 78.77 | N/A | 0.3506 | 0.3569 | 0.3543 | -3.9142 | -0.2950 | 0.9563 | 11.38 | N/A | 0.1105 | 0.0935 | 0.0893 | 0.5064 | 0.2195 |
| XRec [O] | **82.57** | N/A | 0.4193 | **0.4038** | **0.4122** | -3.8035 | **-0.1061** | **1.0000** | **9.60** | N/A | 0.0836 | 0.0920 | 0.0800 | 0.4832 | 0.1780 |
| XRec (w/o prof.) [O] | 81.77 | N/A | **0.4194** | 0.4004 | 0.4106 | -3.8218 | -0.1294 | **1.0000** | **9.60** | N/A | 0.0819 | 0.0955 | **0.0786** | 0.4799 | 0.1803 |
| XRec [R] | N/A | **67.09** | 0.3959 | 0.3990 | 0.3982 | **-3.0586** | -0.1242 | **1.0000** | N/A | 20.48 | 0.0820 | 0.0919 | 0.0790 | 0.3906 | **0.1768** |
| XRec (w/o prof.) [R] | N/A | 64.94 | 0.3982 | 0.3831 | 0.3913 | -3.0855 | -0.2051 | **1.0000** | N/A | 21.86 | 0.0958 | **0.0904** | 0.0837 | **0.3856** | 0.2019 |
| Yelp | | | | | | | | | | | | | | | |
| Att2Seq | 63.91 | N/A | 0.2099 | 0.2658 | 0.2379 | -4.5316 | -0.6707 | 0.7583 | 15.62 | N/A | 0.1583 | 0.1074 | 0.1147 | 0.5616 | 0.247 |
| NRT | 61.94 | N/A | 0.0795 | 0.2225 | 0.1495 | -4.6142 | -0.7913 | 0.2677 | 16.81 | N/A | 0.2293 | 0.1134 | 0.1581 | 0.5612 | 0.2728 |
| PETER | 67.00 | N/A | 0.2102 | 0.2983 | 0.2513 | -4.4100 | -0.5816 | 0.8750 | 15.57 | N/A | 0.3315 | 0.1298 | 0.2230 | 0.5800 | 0.3555 |
| PEPLER | 67.54 | N/A | 0.2920 | 0.3183 | 0.3052 | -4.4563 | -0.3354 | 0.9143 | 14.18 | N/A | 0.1476 | 0.1044 | 0.1050 | 0.5777 | 0.2524 |
| XRec [O] | **74.53** | N/A | **0.3946** | **0.3506** | **0.3730** | -4.3911 | -0.2287 | **1.0000** | 11.45 | N/A | **0.0969** | 0.1048 | **0.0852** | 0.5770 | 0.2322 |
| XRec (w/o prof.) [O] | 71.81 | N/A | 0.3879 | 0.3427 | 0.3657 | -4.4035 | -0.2486 | **1.0000** | 12.71 | N/A | 0.1087 | 0.1072 | 0.0919 | 0.5717 | 0.2272 |
| XRec [R]† | N/A | **53.16** | 0.3236 | 0.3067 | 0.3157 | **-3.6505** | **-0.1900** | **1.0000** | N/A | 19.16 | 0.1076 | 0.1000 | 0.0891 | 0.4374 | **0.2095** |
| XRec (w/o prof.) [R]† | N/A | 51.61 | 0.3920 | 0.3280 | 0.3602 | -3.6912 | -0.3008 | **1.0000** | N/A | 22.14 | 0.1250 | 0.1069 | 0.1008 | 0.4769 | 0.2367 |
| Google-reviews | | | | | | | | | | | | | | | |
| Att2Seq | 61.31 | N/A | 0.3619 | 0.3653 | 0.3636 | -4.2627 | -0.4671 | 0.5070 | 17.47 | N/A | 0.1855 | 0.1247 | 0.1403 | 0.6663 | 0.3198 |
| NRT | 58.27 | N/A | 0.3509 | 0.3495 | 0.3496 | -4.2915 | -0.4838 | 0.2533 | 19.16 | N/A | 0.2176 | 0.1267 | 0.1571 | 0.6620 | 0.3118 |
| PETER | 65.16 | N/A | 0.3892 | 0.3905 | 0.3881 | -4.1527 | -0.3375 | 0.4757 | 17.00 | N/A | 0.2819 | 0.1356 | 0.2005 | 0.6701 | 0.3272 |
| PEPLER | 61.58 | N/A | 0.3373 | 0.3711 | 0.3546 | -4.1744 | -0.2892 | 0.8660 | 17.17 | N/A | 0.1134 | 0.1161 | 0.0999 | 0.6752 | 0.2484 |
| XRec [O] | 69.12 | N/A | **0.4546** | 0.4069 | **0.4311** | -4.1647 | -0.2437 | 0.9993 | 14.24 | N/A | **0.0972** | 0.1163 | 0.0938 | 0.6591 | 0.2452 |
| XRec (w/o prof.) [O] | **69.71** | N/A | 0.4427 | **0.4187** | 0.4310 | -4.1142 | **-0.2026** | 0.9997 | **14.09** | N/A | 0.1180 | 0.1171 | 0.1034 | 0.6465 | **0.2439** |
| XRec [R]† | N/A | **54.41** | 0.4216 | 0.3828 | 0.4026 | -3.5947 | -0.3302 | **1.0000** | N/A | 24.00 | 0.1016 | **0.1106** | **0.0931** | 0.5124 | 0.2461 |
| XRec (w/o prof.) [R]† | N/A | 49.23 | 0.4278 | 0.3916 | 0.4097 | **-3.3692** | -0.3127 | **1.0000** | N/A | 24.82 | 0.1640 | 0.1241 | 0.1283 | 0.5191 | 0.2954 |

XRec. Figure 2(a) shows that Ma et al. (2024) used the GPTScore and BERTScore to make this claim. Our reproduction results in Figure 2(b) show that the removal of these profiles mostly lead to lower mean scores. However, when using the Google-reviews dataset, the mean BERTScore of the experiment with "w/o prof." is slightly higher than the mean BERTScore of the full, unmodified XRec model. This shows that the performance was not necessarily improved by the inclusion of the user and item profiles. As such, we reject claim 3.

**Claim 4: Collaborative information injection.** The final claim states that XRec's performance in explainability and stability metrics is improved by the injection of collaborative information in the transformer layers of XRec's Llama implementation. Again, Figure 2(a) shows that Ma et al. (2024) used GPTScore and BERTScore to make this claim. Figure 2(b) shows that removing the injections consistently led to lower means (lower explainability) and higher standard deviations (lower stability) for the LlamaScore and BERTScore metrics. When training the model without injection on the Amazon-books dataset, we observed anomalies in the generated outputs. Specifically, the model occasionally produced explanations containing numerical sequences, which resulted in broken sentences or an unusable explanation. Examples of these problematic outputs can be found in Appendix B. The lower scores as a result of the removal indicate that performance is improved by injection. Therefore, we validate claim 4.

### 4.2 Ablation Study: Removal of Adapted Embeddings

This subsection relates to RQ 2: "What impact does the removal of the adapted user and item embeddings generated by the MoE have on the generated explanations, in terms of model performance?" To answer this question, we conducted an ablation study that evaluates the impact of the adapted user and item embeddings in XRec. Specifically, this is done by completely removing them from the LLM input. In this setting, the

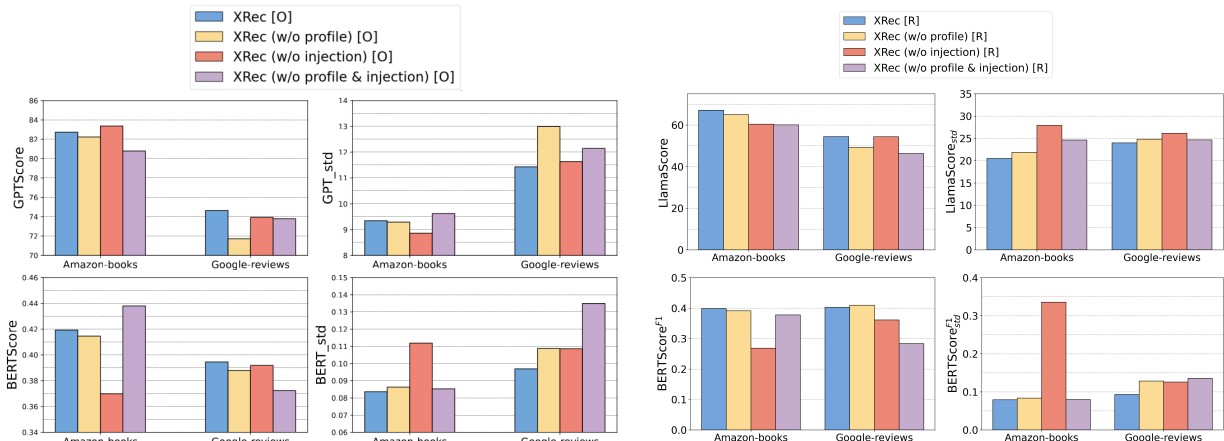

(a) A copy of Figure 3 from Ma et al. (2024), showing the results of their ablation study.

(b) The results of our reproducibility study. Note that the y-axis here is 0-based, as opposed to (a).

Figure 2: Comparison of the results of the ablation study conducted by Ma et al. (2024) and our own. Note the difference in range of the plots. In (a), BERTScore has no superscript, but we assume the original authors to have used BERTScore$^{F1}$ as the F1-score is a more indicative metric than precision or recall.

LLM generates explanations based solely on user and item profiles (see Appendix A for a case study). As we remove the adapted embeddings, no training is needed for this experiment.

The removal of the adapted user and item embeddings generated by the MoE negatively impacts model performance, as reflected in the experimental results (see Table 2). The evaluation metrics were similar to our "w/o injection" reproduction, with a notable 10-point decrease in LlamaScore compared to the unmodified XRec reproduction ("XRec [R]"). As detailed in Appendix A, this removal altered the sentence prefix and structure, leading the model to generate explanations in a more conversational style, or to focus solely on describing the book or establishment. In contrast, explanations that were generated using the adapted embeddings typically followed a structured format, beginning with phrases such as "The user would enjoy" or "The user would buy". To answer research question 2, the MoE module, through its adapted embeddings, plays a role in prompting the LLM to better structure its explanations, and its removal results in degraded performance.

Table 2: Results of our reproduction and extension experiments on the Amazon-books dataset. Comparison in terms of explainability and stability. The superscripts $P$, $R$, and $F1$ indicate precision, recall, and F1-score, respectively. The subscript $std$ denotes the standard deviation of each score. Models marked with † were evaluated using 10% of the test set. The best result for each metric per dataset is highlighted in bold, the second best is underlined.

| Metrics | Explainability ↑ | | | | | | | Stability ↓ | | | | | |
|---|---|---|---|---|---|---|---|---|---|---|---|---|---|
| | Llama | BERT$^P$ | BERT$^R$ | BERT$^{F1}$ | BART | BLEURT | USR | Llama$_{std}$ | BERT$^P_{std}$ | BERT$^R_{std}$ | BERT$^{F1}_{std}$ | BART$_{std}$ | BLEURT$_{std}$ |
| Amazon-books | | | | | | | | | | | | | |
| XRec | **67.09** | 0.3959 | **0.3990** | **0.3982** | **-3.0586** | **-0.1242** | **1.0000** | **20.48** | **0.0820** | 0.0919 | **0.0790** | 0.3906 | **0.1768** |
| XRec w/o profile | 64.94 | **0.3982** | 0.3831 | 0.3913 | -3.0855 | -0.2051 | **1.0000** | 21.86 | 0.0958 | 0.0904 | 0.0837 | 0.3856 | 0.2019 |
| XRec w/o injection | 60.41 | 0.2287 | 0.3126 | 0.2684 | -3.2118 | -0.2625 | 0.9930 | 27.91 | 0.4292 | 0.2409 | 0.3357 | 0.5311 | 0.3657 |
| XRec w/o prof. & inj.† | 60.04 | 0.3915 | 0.3626 | 0.3778 | -3.1284 | -0.3571 | **1.0000** | 24.63 | 0.0905 | **0.0848** | 0.0797 | **0.3603** | 0.2425 |
| XRec w/o embed. | 57.81 | 0.2489 | 0.2912 | 0.2707 | -3.3877 | -0.4551 | 0.9993 | 27.67 | 0.1291 | 0.0916 | 0.1020 | 0.4150 | 0.1948 |
| XRec fixed MoE † | 57.31 | 0.1931 | 0.2950 | 0.2422 | -3.2962 | -0.3125 | **1.0000** | 29.10 | 0.4010 | 0.2090 | 0.3081 | 0.5365 | 0.3888 |

### 4.3 Extension: Random Fixed Mixture of Experts Inputs

After removing the adapted embeddings from the LLM input as described in the previous section, we hypothesized that these embeddings influence not only the meaning of the generated explanations, but also their sentence structure and prefixes. In theory, the adapted embeddings enhance the LLM's explanatory ability by conveying valuable user-item relationship information from the GNN. With this extension, we aim to determine whether these embeddings improve model scores through sentence-structure alignment between generated and ground-truth explanations. Essentially, we aim to investigate if the MoE module is learning to prompt the LLM to produce output sentences of a certain form.

To investigate this and to answer RQ 3: "Do the output embeddings of the GNN provide valuable information to the MoE module for generating better explanations?", we remove the GNN and its output embeddings from the architecture. Instead, we randomly generate fixed user and item embeddings of the same size before training, and use these static embeddings as input for the MoE module for every training example. As a result, the MoE learns a globally optimal setting for the adapted embeddings across the entire dataset, rather than processing unique input embeddings for each example. During inference, we use the same pre-generated embeddings as input to the MoE. If this static setting results in higher evaluation results, it would suggest that the adapted embeddings primarily help the LLM align explanations in sentence structure rather than meaning.

Replacing the GNN embeddings with fixed, pre-generated ones as input to the MoE yielded a LlamaScore similar to the one obtained in Section 4.2. However, this model scored the lowest in most other metrics (see Table 2). Interestingly, the training loss of this model very quickly reached a plateau. Intuitively, this makes sense, as the MoE has to learn to transform only the single set of inputs, as opposed to receiving different embeddings for each training example. Appendix C presents the training loss curve for this model on the Amazon-books dataset, compared to that of the full XRec model. While the performance metrics of these two models differ significantly (see Table 2), their eventual training losses are similar. This might suggest that the training loss is not a very telling metric when comparing these two model architectures.

Some examples of explanations generated using this model are shown in Appendix D. Interestingly, they do show more alignment in sentence structure. This suggests that the embeddings generated by the MoE, used as input for the LLM, do play a role in alignment of the structure of the explanation. However, this effect does not appear to make a positive impact on explainability. Overall, the experiment suggests that using one fixed, general set of MoE-input embeddings for an entire dataset can hinder the model in generating useful explanations.

### 4.4 Consumed Resources and Environmental Impact

In an effort to assess the carbon footprint of our experiments, the $CO_2$ equivalence per experiment was computed as mentioned in Section 3.7. These results are summarized in Table 3. While the MIG nodes appear far more efficient, the training was performed with less samples for both training and generation due to resource constraints. Furthermore, it is unclear how much wattage a MIG instance uses, therefore the calculations assume the full 400 Watt of the A100. In addition, due to the batch size being limited to 1, the GPU had to run for a significantly longer time, resulting in less efficient runtimes. In total, the experiments generated 35.84 kg of $CO_2$ equivalent, which is roughly comparable to driving from Paris to Brussels (336 km) in an average new passenger car (Popov, 2024). As another interesting comparison, we can look at the carbon emissions of the training of Llama-2-7B, which stands at 31.22 tonnes of $CO_2$ equivalent (Touvron et al., 2023). XRec is a very small framework in comparison, building on the language capabilities of this LLM.

Table 3: Training and generation times for various experiments. Training runs marked with † utilized early stopping. All generation on MIG was conducted using 10% of the original test data. Carbon emissions, measured in kilograms of $CO_2$ equivalent, are calculated as the sum of emissions from both training and generation for each experiment.

| Experiment | Training | | Generation | | Emissions |
|---|---|---|---|---|---|
| | GPU | Time | GPU | Time | $CO_2$ equivalent |
| Amazon-books | | | | | |
| XRec | H100 | 9h 54m 58s | H100 | 8h 19m 41s | 6.74 kg |
| XRec w/o profile | H100 | 9h 18m 53s | H100 | 5h 51m 53s | 5.61 kg |
| XRec w/o injecion | H100 | 8h 4m 13s | H100 | 8h 15m 5s | 6.03 kg |
| XRec w/o profile & w/o injection | H100 | 6h 52m 24s† | H100 | 4h 43m 24s | 4.29 kg |
| XRec w/o embeddings | N/A | N/A | MIG | 4h 43m 8s | 0.49 kg |
| XRec with fixed MoE | MIG | 2h 15s† | MIG | 1h 43m 46s | 0.79 kg |
| Yelp | | | | | |
| XRec | H100 | 1h 50m 45s† | MIG | 1h 44m 44s | 1.04 kg |
| XRec w/o profile | H100 | 2h 34m 56s† | MIG | 1h 18m 30s | 1.13 kg |
| Google-reviews | | | | | |
| XRec | H100 | 8h 53m 40s | H100 | 8h 22m 21s | 6.38 kg |
| XRec w/o profile | H100 | 3h 57m 23s† | MIG | 1h 30m 16s | 1.59 kg |
| XRec w/o injecion | MIG | 2h 35m 40s† | MIG | 1h 44m 11s | 0.91 kg |
| XRec w/o profile & w/o injection | MIG | 2h 30m† | MIG | 1h 23m 55s | 0.82 kg |

## 5 Discussion

The XRec framework shows promising performance in generating personalized explanations. However, assessing its effectiveness depends on how well the chosen evaluation metrics capture alignment and explanatory quality. Moreover, alignment between generated and ground truth explanations may inherently be limited as a metric, as it might not fully capture the explanatory value of the generated text. Given that the ground truth explanations were produced by an LLM based on a review, they do not necessarily reflect the actual reasoning behind a recommendation.

When implementing LLamaScore, our replacement of GPTScore, we found the resul ts to be consistently lower than the GPTScores presented in Ma et al. (2024). Table 1 shows a difference of 15 to 20 points between these two metrics. The discrepancy could be attributed to the difference between the two language models and how they interpret and process the task of scoring sentence alignment. This makes it hard to use the LlamaScore metric for evaluating the claims made in the work of Ma et al. (2024). While the other metrics used in this paper do provide clues, a reproduction with the exact GPT model employed by Ma et al. (2024) could be useful.

Moreover, in Section 3.5.2 we theorized that the adapted embeddings (produced by the MoE) boost the scores of the generated results by influencing the sentence structure of the output, particularly by aligning them to look more like the general form of the ground truth explanations (see Appendix A). As shown in Table 2, these adapted embeddings did have the expected effect on sentence structure, but did not result in higher alignment scores. Besides their role in passing on information from the GNN embeddings, their further influence on the generated explanations can be the topic of an interesting follow-up study.

The problematic outputs observed when training and generating using XRec "w/o injection", examples of which are shown in Appendix B, were a surprising result. Ma et al. (2024) state that the injection of embeddings leads to better gradient flow to the MoE. This could explain the difference in results between using XRec with and without injection. However, it is remarkable that adding the MoE-generated embeddings to the prompt is able to destabilize Llama to such an extent.

While being model-agnostic is a perk for explanation models, we feel XRec's biggest limitation lies in its complete independence from the recommender system whose recommendations it aims to explain. Although the user-item interaction graph provides signals to the model through the GNN and custom adapter, this information is inherently limited. We hypothesize that this could become particularly problematic when the

recommender system makes outlier predictions, as XRec may not know how the recommender system came to this unlikely prediction. Such edge cases are often when users would most benefit from clear explanations of the system's decision-making process. Furthermore, the model does not explicitly guarantee the faithfulness of its explanations—whether they accurately reflect the internal decision-making of the recommendation system—a point worth exploring in future research.

In conclusion, we reproduced the results of the XRec paper by Ma et al. (2024) and conducted two additional experiments. Our results demonstrate that XRec effectively generates personalized explanations and that integrating collaborative information enhances the stability of the model's outputs. However, the XRec framework did not consistently surpass all baselines across every metric. Our extended analysis highlighted the critical role of MoE embeddings in shaping explanation structures, highlighting the interaction between collaborative signals and language modeling. Based on our results, we **reject Claim 1** (explainability and stability), **validate Claim 2** (unique explanations), **reject Claim 3** (user and item profiles), and **validate Claim 4** (collaborative information injection). By open-sourcing our implementation and findings, we aim to contribute to ongoing research in utilizing large language models for explainable recommendation systems.

### 5.1 What was easy

The process of starting the reproduction of the work done by Ma et al. (2024) was relatively smooth, as the source code of the XRec framework was provided in a publicly available repository. The code was well-structured, with good instructions and readily available data. Furthermore, the results in Ma et al. (2024) are neatly presented, making for a clear reproduction objective.

### 5.2 What was difficult

The XRec source code lacked certain functionalities necessary for a full reproduction. Notably, the code for evaluation metrics such as BARTScore and BLEURT, as well as the implementations for the "without profile" and ablation studies, were absent and had to be manually implemented. Furthermore, the XRec paper left out critical implementation details, including the number of training epochs, batch size, and the method used to remove low-quality data points from the datasets.

After some debugging, we ran the code without any apparent issues. However, any training and inference in batches unfortunately led to poor results. This led us to train and infer from the model with a batch size of 1, significantly increasing our resource requirements. Due to computational constraints, we generated explanations with only 10% of the test set for some experiments. This means that some of our results may carry less statistical significance than others. Table 3 contains more detailed information about which experiments were affected by this.

### 5.3 Communication with the original authors

We contacted the authors of Ma et al. (2024) to clarify the implementation details of BLEURT and BARTScore. They were very helpful and responded quickly to our questions on GitHub. We used their code as a starting point for our implementation.

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

# A   Examples: Removal of Adapted Embeddings

Like described in Section 3.5.2, we performed an experiment where we generated results without using the adapted input embeddings at all in the LLM input. We compared these results to explanations generated using the full XRec architecture ("w/o injection"). The choice for "w/o injection" here was made for comparison purposes, as in the experiment without embeddings altogether, there is nothing to inject in the first place.

Below are two examples from the Yelp dataset from this case study, each showing the ground truth explanation for the item-user combination, the explanation that was generated using the embeddings (the XRec architecture without injection), and the explanation generated without user and item embeddings.

> **Reference ground truth explanation A**:
> The user would enjoy Don Gabriel for their authentic tacos made with flavorful meats like tinga and carnitas, served with delicious sauces and traditional toppings in a casual setting that adds to the genuine tacqueria experience.

> **Explanation (A) generated using adapted embeddings**:
> The user would enjoy Don Gabriel Restaurant because it offers authentic and flavorful Mexican cuisine, with a focus on quality ingredients and efficient service, creating a casual dining experience that meets their preferences.

> **Explanation (A) generated without using adapted embeddings**:
> Don Gabriel Restaurant offers a unique dining experience with a blend of traditional Mexican flavors and modern twists, using only the freshest and highest-quality ingredients. The cozy atmosphere and attentive service create a welcoming environment for a memorable night out.

> **Reference ground truth explanation B:**
> You would enjoy Grace Meat + Three for its comforting Southern American food, welcoming ordering process, hospitable servers, tasty dishes like Rick's Famous Fried Chicken, and flavorful sides like succotash that perfectly complement the main dishes.

> **Explanation (B) generated using adapted embeddings:**
> The user would enjoy Grace Meat + Three because it offers a variety of flavorful Southern and soul food dishes in a cozy atmosphere with good service and affordable prices, making it a great spot for a satisfying and enjoyable dining experience.

> **Explanation (B) generated without using adapted embeddings:**
> Grace Meat + Three offers a unique, farm-to-table dining experience with seasonal, locally-sourced ingredients, creative flavor combinations, and a cozy atmosphere. Here, customers can enjoy a delicious, satisfying meal with a focus on quality and community.

The adapted embeddings cause an interesting side-effect: they seem to influence the LLM to produce explanations that start with "The user would enjoy".

> **Explanation (C) generated without using adapted embeddings:**
> Sure! Here's why a user might buy "Letting Go (The Garage Series) (Volume 1)" in 50 words or less:
> "Experience the heart-wrenching journey of a young couple's love and loss in this emotionally charged novel. Perfect for fans of John Green and Rainbow Rowell."

This explanation of a recommendation from the Amazon-books dataset shows that the model is conversationally responding to the system prompt (see Appendix E) in the output explanation. This change in response structure occurred often in the experiment without adapted embeddings.

## B    Examples: Generated Explanations Containing Numbers

In some of our experiments, the generated explanations contained some numbers. In some cases this resulted in a broken sentence, whereas in other cases the whole generated explanation string consisted of just numbers. This occurred in the experiments where we reproduced the authors' "w/o injection" results (Section 4.1), and in the experiments where the inputs to the MoE module were fixed (Section 4.3). Four examples of these results are shown below:

> **Generated explanation without injection:**
> 00927/231317 The user would buy the book because it offers a sweet, heartwarming, and emotional story with a strong female protagonist, a complex hero, and a compelling plot centered around a wedding. The book provides a glimpse sense of community, and family, and a touch of redemption, making it a satisfying and enjoyable read.

> **Generated explanation without injection:**
> 76765116082302882220643210782810806060280602028102060202010201020102010101010101010101010101010101 010101010101010101010101010101010

> **Generated explanation using fixed MoE inputs:**
> 44890/user review] 84950/user review for the book, highlighting its captivating and emotional storytelling, as well as the author's ability to weave a story that leaves readers feeling invested in the characters and their journey.

> **Generated explanation using fixed MoE inputs:**
> 1420/40/20/20/40/20/20/20/20/20/20/20/20/20/20/20/20/20/20/20/20/20/20/20/20/20/20/20/20/20/2 0/20/20/20/20/20/20/20/20/20/20

## C    Train Loss

Figure 3 below shows the train loss during training on the Amazon-books dataset of the full, unmodified XRec model, and the train loss during our training of our adaptation model which uses pre-generated, fixed embeddings as input to the MoE module (Section 4.3).

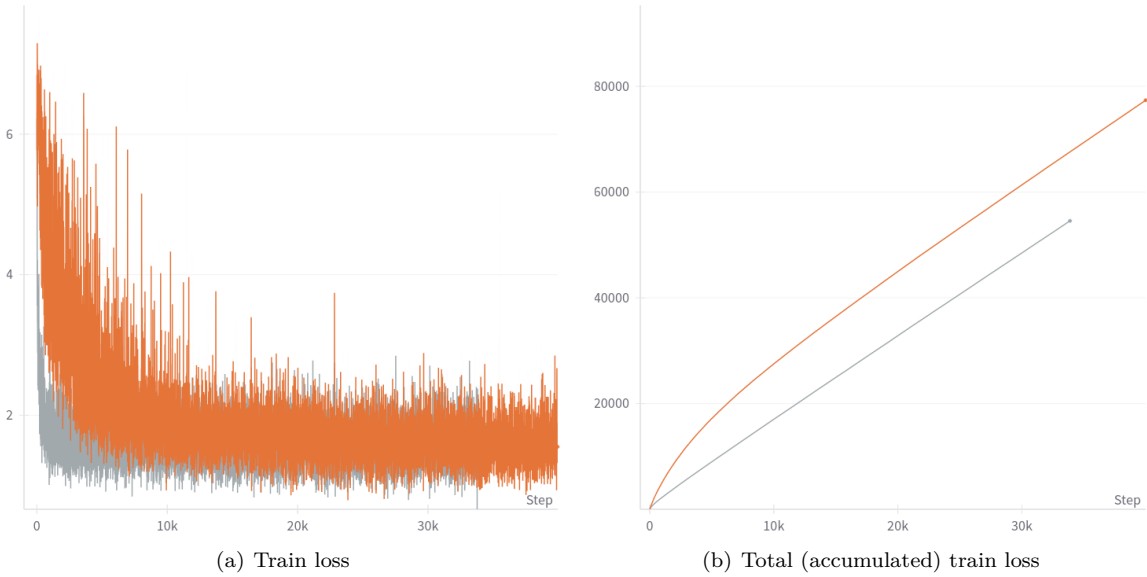

(a) Train loss                                (b) Total (accumulated) train loss

Figure 3: Train loss during training of full XRec model (orange) and model that used fixed embeddings MoE input embeddings (gray). Training is done on the Amazon-books dataset.

# D  Examples: fixed input to MoE

Like theorized in Section 4.3, using the same randomly pre-generated fixed embeddings as inputs to the MoE module generally led to some more alignment in sentence structure between ground-truth and generated explanations, especially in the first few words of the sentences (compare these to the sentences shown in Appendix A).

Below are some examples of this:

> **Reference ground truth explanation D:**
> The user would buy the book because it offers an original and captivating paranormal story with vibrant, well-developed characters, including a strong female lead. The action-packed plot, unexpected twists, and amazing ending make it a must-read for anyone looking for a unique and thrilling paranormal experience.

> **Explanation (D) generated using fixed MoE inputs:**
> The user would buy the book because it offers a unique blend of science fiction and paranormal elements, with a captivating plot involving a group of people with psychic powers, a mysterious prophecy, and a compelling love storyline, all set in a small town with a secret underground world. The book offers an engaging and thrilling storyline that keeps the reader on the edge of their seat, making it a must-read for fans of the genre.

> **Reference ground truth explanation E:**
> The user would buy the book because it features a poignant and gripping story that delves into themes of loss, grief, and the transformative power of art, with well-developed characters and enough twists to keep the reader engaged throughout.

> **Explanation (E) generated using fixed MoE inputs:**
> The user would buy the book because it offers a captivating and emotional journey of a young boy's loss, grief, and survival, intertwined with art, friendship, and the search for identity. The book's themes of loss, regret, and the meaning of life would resonate deeply with the user, making it a compelling and emotional read.

# E System prompt

For scoring the alignment between the generated and ground truth explanation, Ma et al. (2024) prompted GPT-3.5-turbo with the system prompt below. We used this same prompt for generating our alignment scores with Llama-2-7B.

> Score the given explanation against the ground truth on a scale from 0 to 100, focusing on the alignment of meanings rather than the formatting.
> Provide your score as a number and do not provide any other text.

