# OpenReview forum: "Reproducibility Study of "XRec: Large Language Models for Explainable Recommendation""
_TMLR — Withdrawn by Authors_

### Review · Reviewer_wPkN · 2025-03-21

**Summary Of Contributions:**

The paper is a reproducibility study of a recent EMNLP findings paper called "XRec: Large Language Models for Explainable Recommendations". The study uses Llama 3 (instead of ChatGPT as in the original paper). The study also makes some tweaks including modifying the embeddings in MoE modules. Based on the experiments, XRec is proven to be effective but not necessarily outperform all baselines. Further analysis shows the importance of MoR Embeddings. The verification of the claims made by XRec is the following:

Claim 1: Explainability and stability. The claim is rejected due to inconsistent results from the study.
Claim 2: Unique explanations. Validated.
Claim 3: User and item profiles. Rejected due to inconsistency on a dataset.
Claim 4: Collaborative information injection. Validated.

**Audience:**

Yes

**Claims And Evidence:**

No

**Requested Changes:**

Please refer to the detailed reviews. Clear places for improvements are stronger arguments on why XRec, why there might be a nontrival interest from the TMLR community. Stronger arguments on the analysis due to inconsistency of test dataset portions, and the lack of confidence internals (while the combination of the two is even worse).

**Strengths And Weaknesses:**

Strength

The paper is clearly written. The structure is clear. The technical details are clear to the reviewer.

Though the reviewer did not actually run it, the publicly shared code looks good to the reviewer. The repo looks well structured and the files look sensible.

The reproducibility experiments are relatively comprehensive, such as including all major baselines and datasets from the original paper. It also included some abalation study topics in the original paper.

Overall this looks to be a solid reprodubility report for those who might be interested.

Weakness

The motivation of the study is not fully convincing to the reviewer. In the introduction, the paper mentions XRec is a promising work (more about this next) and "The aim of this paper is to reproduce the results presented by Ma etal.". This argument does not well motivate a reproducibility study. There are some discussions scattered in the paper, such the XRec did some pre-processing of the data that is not completely clear.  Also, though the reviewer believes XRec is likely an interesting work, it has <20 citations as the time of reviewing. The reviewer did a google search and was not able to find a lot of discussions of this work. So there's concern in terms of the scope of interest from the TMLR community (where recommender system is an ok, but not focused area already.) The authors may consider making a stronger argument why XRec by putting together the major reasons in the introduction.

Though some findings are surely interesting, the reviewer does not find many surprising findings. This may be partially due to the XRec paper not getting too much attention. A good reference paper could be "Neural Collaborative Filtering vs. Matrix Factorization Revisited", where the authors challenged a heated topic back then.

The conclusions are also less convincing due to some details. For example, the authors used 10% of test set. First, is it even fair to mix numbers using different portions of test set in the same table, such as Table 2? Second, the paper does not provide confidence intervals, making the claims not fully convincing.

As one major extension study topic, the motivation of investigating the influence of user and item embeddings generated by the MoE module is not well motivated - "we aim to investigate if the MoE module is learning to prompt the LLM to produce output sentences of a certain form", which looks vague and not very deep. Also, the extensions look very specific to XRec. Furthermore, the extension study was only on one dataset somehow.

In summary, though the reviewer believes this is an interesting study with some interesting findings, the reviewer is not fully convinced of its sufficient interest to TMLR due to the lack of generalization and some caveats in experimental validations. The reviewer is though open to evaluate potential stronger arguments.

---

### Review · Reviewer_5BWC · 2025-03-25

**Summary Of Contributions:**

The paper reproduces the experiments of the paper on XRec, an LLM model for explainable recommendation. The LLM is changed from GPT-3.5-turbo to Llama 3, with a decrease in performance. There are some additional experiments modifying the input embeddings or deleting the output embeddings. The authors of the original paper made several claims, including explainability and stability (not confirmed), unique explanations (confirmed), the usefulness of the user and item profiles (not confirmed) and the usefulness of injecting collaborative information (confirmed).

**Audience:**

No

**Broader Impact Concerns:**

None.

**Claims And Evidence:**

Yes

**Requested Changes:**

I think the current framework is too far to be published in TMLR, so I am not really suggesting changes. However, the issue of the drop in performance when LLM was changed puzzles me. All the data sets are historical, and that some of the LLM could have been trained on data that is related to the items included, which is sort of cheating in the testing, because in real applications the data is new. I wonder how the particular LLMs potential familiarity with the items could be tested in relation to the performance in the CF systems. This could provide new insights that would be applicable to a larger class of experiments on CFs with LLMs.

**Strengths And Weaknesses:**

It is a well written paper, and the provided code and data could help scientist who are trying to use XRec.

The added knowledge is fairly limited. Some settings are tweaked, but there is no new insights that would be applicable beyond the current setting.

---

### Review · Reviewer_MAXA · 2025-04-22

**Summary Of Contributions:**

This paper presents a reproducibility study of "XRec" (Ma et al., 2024), a framework that injects collaborative information into LLMs for explainable recommendation.

The authors raised a reproducible paper to clarify four claims. They reproduced the original setup and extended it with ablation studies, replacing GNN embeddings, randomly fixing the Mixture of Experts (MoE) module, and reimplementing the evaluation pipeline using open-source models, such as LLamaScore. Experiments support their discussions in terms of the four claims. Details are given in the appendix and the anonymous repository.

Overall, this submission contributes valuable insights regarding the role of the GNN component and the MoE adapter, while also offering practical improvements to reproducibility.

**Audience:**

Yes

**Broader Impact Concerns:**

This paper does not raise any ethical implications.

**Claims And Evidence:**

Yes

**Requested Changes:**

Related to weaknesses, more explicitly and clearly explaining the basis of the reproducing task is required to highlight the contributions.

**Strengths And Weaknesses:**

**Strengths**
- [S1] Comprehensive Reproduction and Extensions: The paper not only replicates the original results but adds value by testing multiple ablation configurations (e.g., GNN and MoE) thereby exploring the architectural assumptions behind XRec, enriching the value of reproducibility papers.
- [S2] Reassessment of Original Claims: The authors revisit four claims from Ma et al. (2024) and provide evidence-based verification or rejection
- [S3] Open-Source Evaluation Framework: By replacing GPTScore with LlamaScore, the authors make the evaluation more accessible and reproducible.

**Weaknesses**
- [W1] Underexplained Diagram and Details: As in the original XRec paper, Figure 1 (architecture diagram) is reused without sufficient re-annotation. The injection mechanism and MoE placement in LLM layers are not clearly described or visually reinforced. Some ablation studies should be explained in Fig. 1 to clarify the importance and highlight the contributions (e.g., compared with a similar paper https://openreview.net/forum?id=cPtqOkxQqH).
- [W2] Evaluation Metric Substitution Raises Concerns: The LlamaScore is introduced as an open alternative to GPTScore, but the authors note a consistent 15–20 point difference between the two. This discrepancy, while acknowledged, is not fully addressed in terms of how it affects the comparability or credibility of conclusions.

---

### Note · Authors · 2025-05-01

**Comment:**

Dear TMLR Action Editor,

We would like to thank you and the reviewers for the insightful feedback on our paper. We appreciate the time and effort the reviewers have dedicated to evaluating our work. Their detailed comments have been invaluable and have helped us better understand the strengths and weaknesses of our submission.

After careful consideration, we have decided that, at this stage, it may not be the best fit for TMLR. Therefore, we have decided to withdraw our paper from the review process.

We sincerely hope to have the opportunity to submit to TMLR again in the future, and once again, we thank the reviewers for their feedback.

Kind regards,

The Authors

**Withdrawal Confirmation:**

I have read and agree with the venue's withdrawal policy on behalf of myself and my co-authors.